# Additions to *Macgarvieomyces* in Iran: Morphological and Phylogenetic Analyses Reveal Six New Species

**DOI:** 10.3390/jof11070489

**Published:** 2025-06-27

**Authors:** Abdollah Ahmadpour, Youbert Ghosta, Fatemeh Alavi, Zahra Alavi, Esmaeil Hashemlou, Jaturong Kumla, Samantha C. Karunarathna, Nakarin Suwannarach

**Affiliations:** 1Higher Education Center of Shahid Bakeri, Urmia University, Miyandoab 59781–59111, Iran; 2Department of Plant Protection, Faculty of Agriculture, Urmia University, Urmia 57561–51818, Iran; y.ghoosta@urmia.ac.ir (Y.G.); fatemeh.alavi0211@gmail.com (F.A.); alavizahra996@gmail.com (Z.A.); 3Department of Plant Protection, College of Agriculture and Natural Resources, University of Tehran, Karaj 31587-77871, Iran; ehashemlou@gmail.com; 4Office of Research Administration, Chiang Mai University, Chiang Mai 50200, Thailand; jaturong_yai@hotmail.com; 5Center of Excellence in Microbial Diversity and Sustainable Utilization, Chiang Mai University, Chiang Mai 50200, Thailand; 6Center for Yunnan Plateau Biological Resources Protection and Utilization, College of Biology and Food Engineering, Qujing Normal University, Qujing 655011, China; samanthakarunarathna@gmail.com

**Keywords:** *Magnaporthales*, morphology, new taxa, phylogeny, *Pyriculariaceae*, six new species, taxonomy

## Abstract

The genus *Macgarvieomyces* (*Magnaporthales*, *Sordariomycetes*, *Ascomycota*) currently includes three species, which are associated with leaf spots on plants belonging to the *Cyperaceae* and *Juncaceae* families and are known only in Europe and New Zealand. During a comprehensive survey conducted between 2020 and 2022 targeting host plants from these families across various regions of Iran, six novel species of *Macgarvieomyces*—*M. caspica*, *M. cyperi*, *M. junci-acuti*, *M. juncigenus*, *M. salkadehensis*, and *M. schoeni*—were uncovered. These species were identified based on detailed morphological characterizations and multi-locus phylogenetic analyses using ITS-rDNA, *RPB1*, *ACT*, and *CAL* gene regions. This study provides thorough descriptions and illustrations of the new taxa, including information on their morphology, ecological preferences, and geographic distribution. The phylogenetic relationships among the species are also discussed. This work significantly enhances the known diversity of *Macgarvieomyces* associated with *Cyperaceae* and *Juncaceae*, expands their geographic distribution, and underscores the value of integrating morphological and molecular data in fungal taxonomy; accordingly, the findings of this study lay the groundwork for future ecological and evolutionary studies of this genus.

## 1. Introduction

The order *Magnaporthales* (*Sordariomycetes*, *Ascomycota*), which was originally established to accommodate the *Magnaporthaceae* family according to the combined phylogenetic data of the small and large subunits of ribosomal DNA (SSU and LSU) and morphological features [1], comprises a diverse group of fungi that includes important and destructive plant pathogens, mainly on cereals and grasses—endophytes and saprophytes—associated with submerged wood and dead grasses [2,3,4,5]. Historically, the taxonomy of *Magnaporthales* was based on morphological characteristics, such as the structure of perithecia, asci, and ascospores, as well as the morphological characteristics of the asexual morphs [2,6]. However, morphological traits exhibit high levels of convergence and plasticity among different lineages and have been proven to be insufficient for taxonomic resolution. Consequently, an integrative approach combining morphology, ecology, and multi-locus phylogeny has been adopted for stable classification [2,5]. At present, multi-locus phylogenetic analyses utilizing the large subunit of ribosomal DNA (LSU), the internal transcribed spacer (ITS) region, parts of the largest subunit of RNA polymerase II (*RPB1*), actin (*ACT*), calmodulin (*CAL*), translation elongation factor 1-alpha (*TEF1*), and minichromosome maintenance complex component 7 (*MCM7*) gene sequences have been used to delimit the ordinal boundaries and confirm the monophyly of the order *Magnaporthales.* These studies have resolved the order into six families: *Ceratosphaeriaceae*, *Magnaporthaceae*, *Ophioceraceae*, *Plagiosphaeraceae*, *Pseudohalonectriaceae*, and *Pyriculariaceae*. These comprise 52 genera and 317 species [2,5,7,8,9,10,11].

The family *Pyriculariaceae* was established by Klaubauf et al. [2], with *Pyricularia* as the type of genus, according to the combination of morphological features and phylogenetic data derived from LSU, ITS, *CAL*, *ACT*, and *RPB1* sequences. At present, 13 genera—*Bambusicularia*, *Barretomyces*, *Bipyricularia*, *Macgarvieomyces*, *Neocordana*, *Neopyricularia*, *Nothopyricularia*, *Proxipyricularia*, *Pseudopyricularia*, *Pyricularia*, *Pyriculariomyces*, *Utrechtiana*, and *Xenopyricularia*—are recognized in *Pyriculariaceae*. These genera are primarily characterized by the production of denticulate conidiogenous cells and ellipsoid, obclavate, or pyriform, septate conidia (pyricularia-like conidia) [2,5,7,8,9,12]. Most species within the *Pyriculariaceae* are ecologically characterized by their pathogenicity on cereals and other monocotyledonous plants. A prominent example is *Pyricularia oryzae*, the pathogen responsible for rice blast disease, which is associated with yield losses reaching up to 30% on a global scale [13,14]. The genus *Macgarvieomyces* was introduced to accommodate two species, *Macgarvieomyces borealis* and *M. juncicola*, which were previously classified under *Pyricularia* and were isolated from *Juncus effusus* [2]. The third species, *M. luzulae* (syn. *Pyricularia luzulae*), isolated from *Luzula* spp. (*Juncaceae*), was later included in the genus, bringing the total to three species [9]. Species of *Macgarvieomyces* have been reported from leaf spots on plants in the *Cyperaceae* and *Juncaceae* families in Europe and New Zealand [2,9,15].

Iran hosts remarkably diverse ecosystems, including temperate forests, montane grasslands, arid steppe, and semi-desert biomes with diverse vegetation. In our ongoing studies of fungi from plants in the families *Cyperaceae* and *Juncaceae*, we focused our survey on *Macgarvieomyces* isolates collected in Iran between 2020 and 2022, gathering specimens from a wide array of host plants exhibiting leaf and stem spots and blights to explore the species diversity of this genus. This study aimed to identify and describe *Macgarvieomyces* species based on combined morphological characteristics and multigene phylogenetic analyses using ITS, *RPB1*, *ACT*, and *CAL* sequences.

## 2. Materials and Methods

### 2.1. Sample Collection and Isolation of Fungi

Leaf and culm samples exhibiting brown lesions and blight symptoms were collected from various wetland plants belonging to the families of *Cyperaceae* and *Juncaceae* across six Provinces in Iran: Ardebil, Golestan, Guilan, Mazandaran, Tehran, and West Azarbaijan, between 2020 and 2022 (Figure 1). The samples were labelled [16], stored at low temperatures, and transported to the laboratory. Fungal isolation, purification, and preservation followed the methods described by Ahmadpour et al. [17,18]. The pure cultures of all identified isolates were deposited in the fungal culture collections at the Iranian Research Institute of Plant Protection (IRAN) and Urmia University (FCCUU).

### 2.2. Morphological Examination

Mycelial discs (5 mm in diameter) were excised from the actively growing edges of seven-day-old cultures and transferred to fresh culture media, including Potato Dextrose Agar (PDA; 39 g/L, Merck, Darmstadt, Germany), Potato Carrot Agar (PCA; composed of 20 g white potato, 20 g carrot, 20 g agar, and 1000 mL distilled water), and Malt Extract Agar (MEA; 50 g/L, Quelab, Montreal, QC, Canada). The culture plates were incubated in darkness at 25 °C for periods of 7 and 14 days. Colony characteristics, including colour, growth pattern, and diameter, were observed and documented. Colony colours were determined using Rayner’s [19] colour charts. The morphological characteristics were examined and recorded from 10–14-day-old cultures on synthetic nutrient-poor agar medium (SNA) (KH_2_PO_4_ 1.0 g, KNO_3_ 1.0 g, MgSO_4_ + 7H_2_O 0.5 g, KCl 0.5 g, glucose 0.2 g, sucrose 0.2 g, agar 20 g, distilled water 1000 mL) [20] supplemented with autoclaved barley seeds and incubated at 23–25 °C under 12 h dark/12 h near-ultraviolet light for 10–14 days [2]. Microscopic examination and measurements of hyphae, conidiophores, conidia, hyphopodia, and chlamydospores were performed from fungal slides mounted in clear lactic acid or lactophenol cotton blue solutions. For each morphological structure, 30 to 50 measurements were obtained, and images were captured using an Olympus AX70 microscope equipped with differential interference contrast (DIC) illumination (Olympus Optical Co., Ltd., Tokyo, Japan). Images were processed using Adobe Photoshop 2020 version 2.10.8 (Adobe Inc., San Jose, CA, USA). Newly described taxa were registered in the MycoBank database (www.mycobank.org; accessed 20 May 2025) [21].

### 2.3. DNA Extraction, PCR Amplification, and Sequencing

Total genomic DNA was isolated from the mycelial biomass of each isolate, obtained from 10-day-old cultures grown on PDA, using the protocol outlined by Ahmadpour et al. [18]. The ITS region, *RPB1*, *ACT*, and *CAL* genes were amplified using the primer pairs ITS1/ITS4 [22], RPB1F/RPB1R [2], ACT-512F/ACT-783R [23], and CAL-228F/CAL-737R [23], respectively. PCR amplification was conducted using a SimpliAmp™ Thermal Cycler (Applied Biosystems™, Thermo Fisher Scientific Corp., Waltham, MA, USA) in a final reaction volume of 30 μL. Each reaction mixture comprised 0.4 μM of each primer, 10 μL of 2X Master Mix containing Taq DNA polymerase and 2 mM MgCl_2_ (Ampliqon, Odense, Denmark), and approximately 10 ng of genomic DNA. PCR amplification was carried out under the following thermal cycling conditions: an initial denaturation at 95 °C for 5 min, followed by 35 cycles consisting of denaturation at 95 °C for 45 s, annealing at 62–57 °C for ITS, *ACT*, and *CAL* regions, and 57–52 °C for *RPB1* (with a gradual decrease of 0.5 °C per cycle during the first 10 cycles), for 45 s, and extension at 72 °C for 45 s. A final extension step was performed at 72 °C for 7 min. PCR amplicons were visualized on a 1% agarose gel stained with FluoroVue™ Nucleic Acid Gel Stain (SMOBIO Technology Inc., Hsinchu, China), and fragment sizes were estimated using the FluoroBand™ 100 bp + 3K Fluorescent DNA Ladder (SMOBIO Technology Inc., China). Macrogen Corp. (Seoul, Republic of Korea) cleaned and sequenced the amplified products using the same primer sets employed for PCR amplification. The new sequences were submitted to GenBank (Table 1).

### 2.4. Molecular Phylogeny

Preliminary identification of the isolates was conducted by comparing newly generated ITS, *RPB1*, *ACT*, and *CAL* sequences using the NCBI Basic Local Alignment Search Tool (BLAST) (www.ncbi.nlm.nih.gov/blast/; accessed 30 April 2025). Subsequently, pairwise sequence comparisons were carried out between the putative novel species and their closest relatives using the same platform. Reference sequences from type or representative species were retrieved from GenBank (Table 1), based on recent studies by Klaubauf et al. [2], Feng et al. [5], and Marin-Felix et al. [9], and incorporated into the analyses. A multi-locus phylogenetic analysis was performed using a concatenated dataset consisting of four genetic loci (ITS, *RPB1*, *ACT*, and *CAL*). Sequence alignments were generated with the online platform MAFFT version 7 (https://mafft.cbrc.jp/alignment/server/; accessed 20 April 2025) [24]. The most appropriate nucleotide substitution models were selected based on the Akaike Information Criterion (AIC) implemented in MrModeltest version 2.3 [25]. Maximum likelihood (ML), Bayesian phylogenetic inference (BI), and maximum parsimony (MP) analyses were performed using the CIPRES Science Gateway portal version 3.3 (https://www.phylo.org/; accessed 20 April 2025) [26]. ML analysis was conducted with RAxML-HPC BlackBox v. 8.2.12 employing the GTR + GAMMA model and 1000 bootstrap replicates [27]. Bayesian inference was carried out using MrBayes on ACCESS v. 3.2.7a, applying the Markov Chain Monte Carlo (MCMC) method with four chains run for 1,000,000 generations, sampling every 1000 generations, and discarding the first 25% as burn-in [28]. Parsimony analysis was performed in PAUP on ACCESS v. 4.a168 using a heuristic search strategy and tree bisection and reconnection (TBR) branch swapping, with 1000 bootstrap replicates [29]. Descriptive statistics for the parsimony trees, including Tree Length (TL), Consistency Index (CI), Retention Index (RI), and Homoplasy Index (HI), were also calculated. In all phylogenetic analyses, *Magnaporthiopsis incrustans* (M35) and *Ma. poae* (ATCC 64411) were employed as outgroup taxa [5]. Phylogenetic trees were visualized using FigTree version 1.4.4 [30] and subsequently refined with Adobe Illustrator^®^ CC 2021 (Adobe Inc., San Jose, CA, USA) for graphical presentation.

### 2.5. Genealogical Concordance Phylogenetic Species Recognition Analysis

Genealogical Concordance Phylogenetic Species Recognition (GCPSR) was applied to detect notable recombination events among phylogenetically close species [31]. A concatenated dataset comprising four loci (ITS, *RPB1*, *ACT*, and *CAL*) was analysed using SplitsTree version 5 software, applying the pairwise Homoplasy Index (PHI or Φw) test [32,33]. The phylogenetic relationships between the newly identified species and their closely related taxa were visualized by generating split graphs from the concatenated dataset, employing the LogDet transformation and split decomposition methods. A PHI test value below 0.05 (Φw < 0.05) signifies the occurrence of significant recombination within the dataset.

## 3. Results

### 3.1. Phylogenetic Analyses

In this study, 95 isolates were obtained from various plants in the *Cyperaceae* and *Juncaceae* families. All isolates were examined based on their morphology, and 18 representative isolates were then selected according to different plant hosts for phylogenetic study. A total of 57 ITS, 56 *RPB1*, 56 *ACT*, and 48 *CAL* sequences were subjected to multiple sequence alignment (nucleotides + gaps), resulting in 517-, 735-, 513-, and 670-character datasets, respectively. The four-gene sequence combination for a total of 57 strains consisted of 2435 characters; of these, 1185 were constant, 129 were variable and parsimony-uninformative, and 1121 were parsimony-informative (Table 2). The most parsimonious tree yielded the following values: TL = 4826, CI = 0.474, RI = 0.760, and HI = 0.526. The MrModeltest findings suggested the GTR+I+G, GTR+I+G, HKY+I+G, and HKY+I+G models for ITS, *RPB1*, *ACT*, and *CAL* datasets, respectively (Table 2). Phylogenetic information and substitution models for each dataset is summarized in Table 2. The ML, MP, and BI phylogenetic analyses yielded congruent tree topologies with no significant conflicts observed. The combined dataset analysis conducted using RAxML produced the best-scoring tree (Figure 2), with a final ML optimization likelihood value of –23,503.286101. The estimated nucleotide base frequencies were A = 0.251726, C = 0.283076, G = 0.255946, and T = 0.209252. The substitution rates were calculated as follows: AC = 1.085584, AG = 2.836763, AT = 1.235728, CG = 0.790339, CT = 4.384725, and GT = 1.000000. The gamma distribution shape parameter (α) was estimated at 1.489603. All 18 studied isolates were clustered into the genus *Macgarvieomyces* (Figure 2). Based on morphological characteristics, multi-locus phylogenetic analyses (ITS, *RPB1*, *ACT*, and *CAL*), and the pairwise Homoplasy Index (PHI or Φw) test, six new *Macgarvieomyces* species *viz. Macgarvieomyces caspica*, *M. cyperi*, *M. junci-acuti*, *M. juncigenus*, *M. salkadehensis*, and *M. schoeni* were identified. Comprehensive morphological descriptions and illustrations were provided for all species, alongside detailed discussions of their habitat, distribution, and phylogenetic relationships with other *Macgarvieomyces* taxa.

### 3.2. Taxonomy

#### 3.2.1. *Macgarvieomyces caspica* A. Ahmadpour, Y. Ghosta, F. Alavi, Z. Alavi, and E. Hashemlou sp. nov. (Figure 3)

MycoBank No. 859375

Etymology: The name refers to the Caspian Sea, where the holotype was collected.

Typification: Iran, Golestan Province, Torkaman County, Qareh Su Village, isolated from the culms of *Juncus* sp. (*Juncaceae*, *Poales*), 7 August 2021, *A. Ahmadpour* (holotype IRAN 18495F, ex-type culture IRAN 5071C).

*Asexual morph* on SNA medium with sterile barley seed: *Mycelium* consisting of smooth, hyaline, branched, septate hyphae, 1–2 μm diam. *Conidiophores* semi–macronematous, solitary, erect, straight–flexuous, unbranched, thick-walled, pale–medium brown, 1–2-septate, lacking a swollen base, 45–125 × 4–5 µm ($\bar{x}$ = 77 × 4.5 μm, *n* = 50). *Conidiogenous cells* integrated, terminal, rarely intercalary, pale brown, smooth-walled, forming a rachis with several sympodially protruding flat-tipped denticles, 1–2 × 1–1.5 μm. *Conidia* solitary, obclavate–pyriform–fusoid, hyaline at first and becoming pale brown with age, smooth, granular, guttulate, 1-septate, not or slightly constricted at septum, apex obtusely rounded, base tapering towards a protruding hilum, 1–1.5 μm diam., not thickened, not darkened, 22–27 × 6–7 µm ($\bar{x}$ = 24.5 × 6.5 μm, *n* = 50). *Hyphopodia*, *sexual morph*, *microconidiation*, and *chlamydospores* were not observed.

Culture characteristics: Colony on PDA reaching up to 35 and 55 mm diam. after 7 and 14 days at 25 °C in the dark, respectively; flat, circular, margin regular, white with buff centre, reverse white with buff centre. Colony on PCA reaching up to 35 and 53 mm diam. after 7 and 14 days at 25 °C, respectively; flat, circular, margin regular, white with buff centre and white aerial mycelium, reverse white with buff centre. Colony on MEA reaching up to 40 and 54 mm diam. after 7 and 14 days at 25 °C, respectively; flat, circular, margin entire, transparent, velvety, isabelline, or pale luteous with white aerial mycelium at the centre, reverse ochreous–pale luteous towards the edge.

Additional specimen examined. Iran, Golestan Province, Torkaman County, Qareh Su Village, isolated from the culms of *Juncus* sp. (*Juncaceae*, *Poales*), 7 August 2021, *A. Ahmadpour* (culture FCCUU 1956).

Notes: Phylogenetic analyses indicate that *Macgarvieomyces caspica* is closely affiliated with *M. junci-acuti* and *M. schoeni*, supported by high confidence values (MLBS/MPBS/BIPP = 100/100/1.0) (Figure 2). A comparison of nucleotide differences in ITS, *RPB1*, *ACT*, and *CAL* indicates that *M. caspica* (IRAN 5071C) differs from *M. junci-acuti* (IRAN 4233C) by 2/434 bp (0.69%) in ITS, 10/693 bp (1.44%) in *RPB1*, 15/300 bp (5%, with five gaps (1%)) in *ACT*, and 23/460 bp (5%) in *CAL* and from *M. schoeni* (IRAN 5074C) by 3/479 bp (0.62%) in ITS, 3/688 bp (0.43%) in *RPB1*, 5/303 bp (1.65%, with one gap (0%)) in *ACT*, and 3/473 bp (0.63%, with two gaps (0%)) in *CAL*. The PHI analysis results indicated that *M. caspica* does not exhibit significant genetic recombination with its closely related species (Φw > 0.05, Figure 4). *Macgarvieomyces caspica* can be differentiated from *M. junci-acuti* by producing shorter conidiophores (45–125 µm vs. 80–150 µm in *M. junci-acuti*), shorter and slightly wider conidia (22–27 × 6–7 µm vs. 25–33 × 5–6 μm in *M. junci-acuti*), and conidial shape (obclavate to pyriform vs. narrowly obclavate in *M. junci-acuti*), and from *M. schoeni* by shorter and slightly narrower conidia (22–27 × 6–7 µm vs. 26–32 × 4–5 μm in *M. schoeni*), and absence of hyphopodia.
Figure 3*Macgarvieomyces caspica* (IRAN 5071C, ex-type). (**a**,**b**) Symptoms on the culms of *Juncus* sp.; (**c**–**e**) colony on PDA (**c**), PCA (**d**), and MEA (**e**) after 14 days; (**f**–**i**) sporulation pattern on SNA medium ((**f**) = 10×, (**g**) = 20×, (**h**,**i**) = 40×); (**j**–**o**) conidiophores and conidia. Scale bars: (**j**–**o**) = 10 μm.
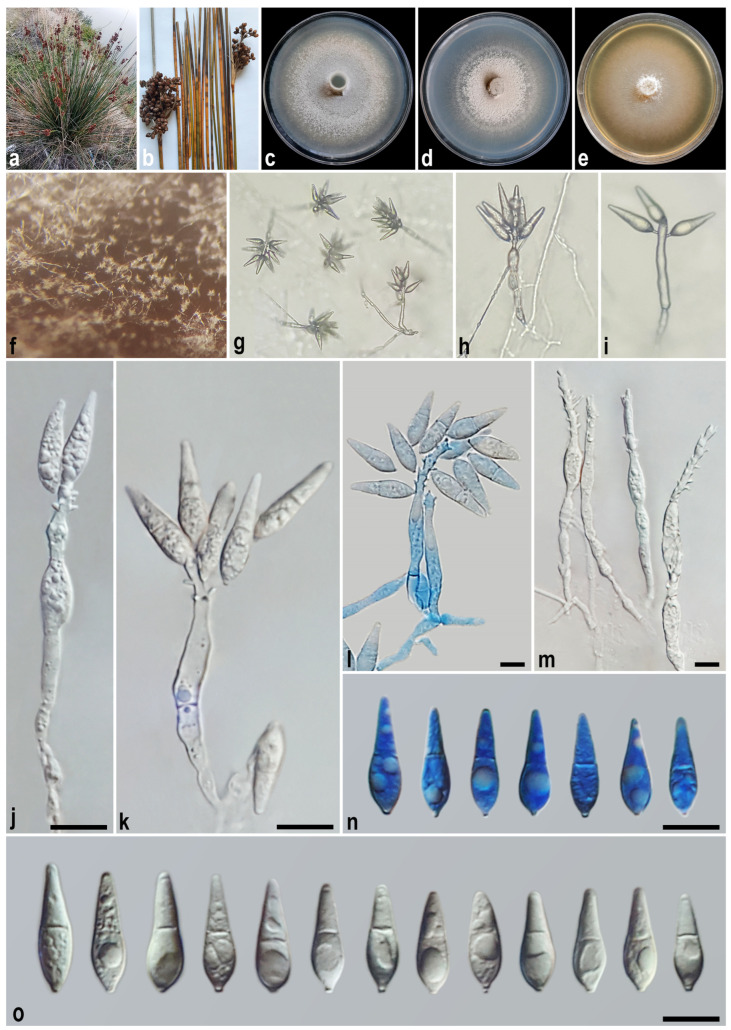

Figure 4Split graph illustrating the pairwise Homoplasy Index (PHI) test results for the newly identified species and their closely related taxa, generated using LogDet transformation and split decomposition methods. A PHI test *p*-value (Φw) ≤ 0.05 signifies significant recombination among the aligned isolates. Newly described taxa are highlighted in bold blue, with the PHI test value and scale bars presented in the lower right corner.
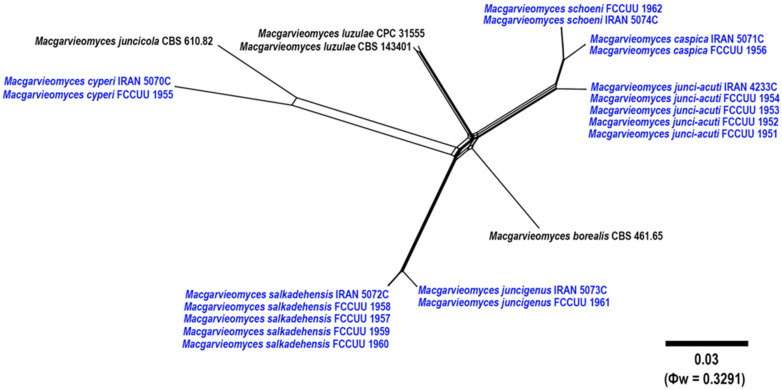


#### 3.2.2. *Macgarvieomyces cyperi* A. Ahmadpour, Y. Ghosta, F. Alavi, Z. Alavi, and E. Hashemlou, sp. nov. (Figure 5)

MycoBank No. 859376

Etymology: Named after the host genus, *Cyperus*, from which the holotype was collected.

Typification: Iran, Guilan Province, Fuman County, Masuleh City, isolated from the leaves and culms of *Cyperus* sp. (*Cyperaceae*, *Poales*), 6 August 2022, *A. Ahmadpour* (holotype IRAN 18494F, ex-type culture IRAN 5070C).

Description: *Asexual morph* on SNA medium with sterile barley seed: *Mycelium* consisting of smooth, hyaline, branched, septate hyphae, 1.2–2 μm diam. *Conidiophores*, semi–macronematous, solitary, erect, straight–flexuous, smooth, unbranched, medium brown–dark brown, 1–3-septate, tapering and paler towards the apex, mostly simple, occasionally swollen at the base, 50–125 × 4–5 µm ($\bar{x}$ = 85 × 4.5 μm, *n* = 50). *Conidiogenous cells* integrated, terminal, rarely intercalary, pale brown to brown, smooth, forming a rachis with several sympodially protruding denticles, 1–1.5 × 1–1.2 μm. *Conidia* solitary, narrowly obclavate, hyaline, becoming pale brown with age, smooth, granular, guttulate, 1-septate, not or slightly constricted at septum, apex obtusely rounded, base tapering to a protruding hilum, 1–1.2 μm diam., not thickened, not darkened, 19–25 × 4–6 µm ($\bar{x}$ = 22 × 5 μm, *n* = 50). *Hyphopodia*, *sexual morph*, *microconidiation*, and *chlamydospores* were not observed.

Culture characteristics: Colony on PDA reaching up to 34 and 50 mm diam. after 7 and 14 days at 25 °C in the dark, respectively; flat, circular, margin regular, white with buff centre and white aerial mycelium, reverse white with buff centre. Colony on PCA reaching up to 35 and 52 mm diam. after 7 and 14 days at 25 °C, respectively; flat, circular, margin regular, velvety or wool-like texture, white with pale olivaceous grey mycelium at the centre, reverse white with pale olivaceous grey centre. Colony on MEA reaching up to 35 and 52 mm diam. after 7 and 14 days at 25 °C, respectively; flat, circular, margin entire, velvety, isabelline, or pale luteous with white aerial mycelium at the centre, reverse ochreous–pale luteous towards the edge.

Additional specimen examined. Iran, Guilan Province, Asalem County, isolated from the leaves and culms of *Cyperus* sp. (*Cyperaceae*, *Poales*), 6 June 2022, *A. Ahmadpour* (culture FCCUU 1955).

Notes: *Macgarvieomyces cyperi* is phylogenetically closely allied with *M. juncicola*, supported by strong bootstrap and posterior probability values (MLBS/MPBS/BIPP = 100/100/1.0) (Figure 2). A comparison of nucleotide differences in ITS, *RPB1*, *ACT*, and *CAL* indicates that *M. cyperi* (IRAN 5070C) differs from *M. juncicola* (CBS 610.82) by 12/453 bp (2.64%, with three gaps (0%)) in ITS, 37/689 bp (5.37%, with one gap (0%)) in *RPB1*, 43/292 bp (14.72%, with 14 gaps (4%)) in *ACT*, and 52/455 bp (11.42%, with four gaps (0%)) in *CAL*. The PHI analysis results confirmed that *M. cyperi* does not exhibit significant genetic recombination with its closely related species (Φw > 0.05, Figure 4). Morphologically, *M. cyperi* can be differentiated from *M. juncicola* by shorter conidiophores (up to 125 vs. up to 200 μm in *M. juncicola*), shorter conidia (19–25 vs. 17–32 μm) in *M. juncicola*, and the formation of chlamydospores in intercalary chains in *M. juncicola* [2].
Figure 5*Macgarvieomyces cyperi* (IRAN 5070C, ex-type). (**a**) Symptoms on the leaves of *Cyperus* sp.; (**b**–**d**) colony on PDA (**b**), PCA (**c**), and MEA (**d**) after 14 days; (**e**,**f**) sporulation pattern on SNA medium (20×); (**g**–**n**) conidiophores and conidia. Scale bars: (**g**–**n**) = 10 μm.
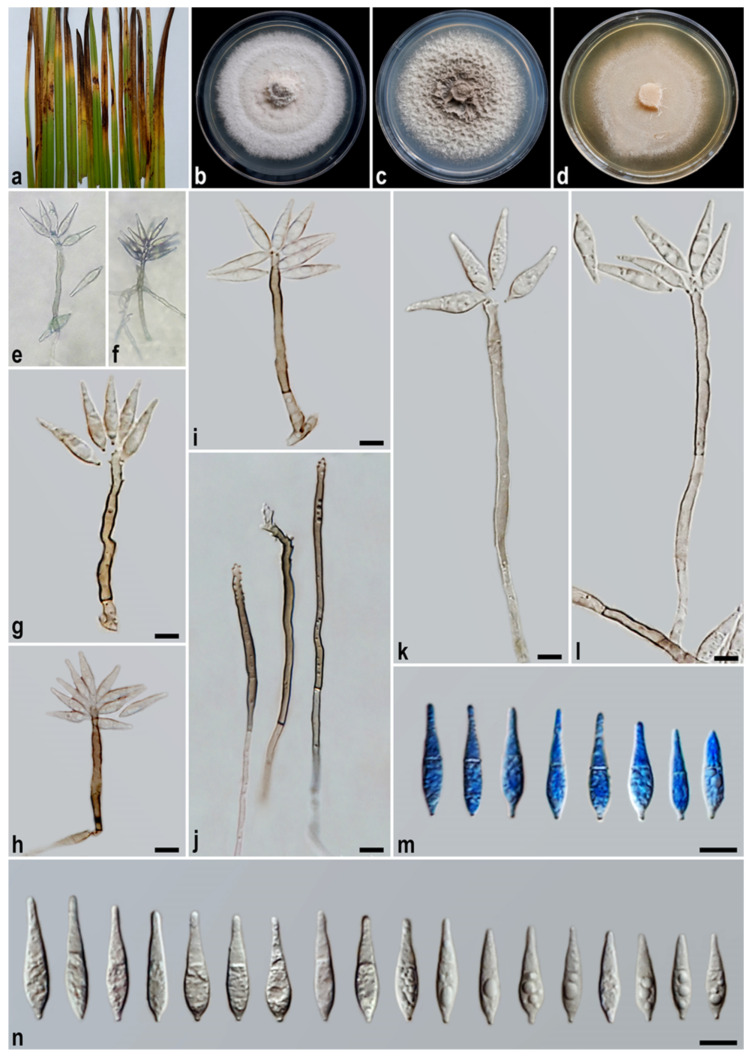


#### 3.2.3. *Macgarvieomyces junci-acuti* A. Ahmadpour, Y. Ghosta, F. Alavi, Z. Alavi, and E. Hashemlou, sp. nov. (Figure 6)

MycoBank No. 859377

Etymology: Named after the host genus, *Juncus acutus*, from which the holotype was collected.

Typification: Iran, West Azarbaijan Province, Urmia County, Baran Duz Village, isolated from the culms of *Juncus acutus* (*Juncaceae*, *Poales*), 31 August 2020, *A. Ahmadpour* (holotype IRAN 18105F, ex-type culture IRAN 4233C).

Description: *Asexual morph* on SNA medium with sterile barley seed: *Mycelium* consisting of smooth, hyaline, branched, septate hyphae, 1–2 μm diam. *Conidiophores* semi–macronematous, solitary, erect, straight–flexuous, smooth, mostly unbranched, occasionally branched, hyaline–pale brown, 1–4-septate, thick-walled near the base, 80–150 × 4–5 µm ($\bar{x}$ = 103 × 4.5 μm, *n* = 50). *Conidiogenous cells* integrated, terminal, rarely intercalary, hyaline–pale brown, smooth, forming a rachis with several sympodially protruding flat-tipped denticles, 1–2 × 1–1.5 μm diam. *Conidia* solitary, obclavate–narrowly obclavate, hyaline, becoming pale brown with age, smooth, granular, guttulate, 1-septate, not or slightly constricted at septum, apex obtusely rounded, base tapering to a protruding hilum, 1–1.5 μm diam., not thickened, not darkened, 25–33 × 5–6 µm ($\bar{x}$ = 28.5 × 5.5 μm, *n* = 50). *Hyphopodia* commonly formed, elongated, dome-shaped–lobulate, brown–dark brown, smooth, 10–15 × 5–6 µm. *Sexual morph*, *microconidiation*, and *chlamydospores* were not observed.

Culture characteristics: Colony on PDA reaching up to 25 and 49 mm diam. after 7 and 14 days at 25 °C in the dark, respectively; flat, circular, margin entire, transparent, velvety, isabelline, or pale luteous, reverse ochreous–pale luteous towards the edge. Colony on PCA reaching up to 22 and 45 mm diam. after 7 and 14 days at 25 °C, respectively; flat, circular, margin regular, white–grey with white aerial mycelium, reverse grey at the centre and hyaline at the margin. Colony on MEA reaching up to 20 and 35 mm diam. after 7 and 14 days at 25 °C, respectively; flat, circular, margin entire, cottony appearance, white with white aerial mycelium, reverse white–hyaline towards the edge.

Additional specimens examined: Iran, Golestan Province, Torkaman County, Qareh Su Village, isolated from the culms of *Juncus* sp. (*Juncaceae*, *Poales*), 3 November 2021, *A. Ahmadpour* (culture FCCUU 1951); Iran, Mazandaran Province, Sari County, Qajar Kheyl Village, isolated from the culms of *Juncus* sp. (*Juncaceae*, *Poales*), 2 November 2021, *A. Ahmadpour* (culture FCCUU 1952); Iran, Mazandaran Province, Sari County, Farahabad City, isolated from the culms of *Juncus* sp. (*Juncaceae*, *Poales*), 2 November 2021, *A. Ahmadpour* (culture FCCUU 1953); Iran, West Azarbaijan Province, Khoy County, Salkadeh Village, isolated from the culms of *Schoenus* sp. (*Cyperaceae*, *Poales*), 5 October 2020, *A. Ahmadpour* (culture FCCUU 1954).

Notes: Isolates of *Macgarvieomyces junci-acuti* formed a distinct clade with strong support values, including 100% maximum likelihood (ML) bootstrap, 100% maximum parsimony (MP) bootstrap, and a Bayesian posterior probability (BI) of 1.0, and were resolved as sister taxa to *M. schoeni* and *M. caspica* (Figure 2). A comparison of nucleotide differences in ITS, *RPB1*, *ACT*, and *CAL* indicates that *M. junci-acuti* (IRAN 4233C) differs from *M. caspica* (IRAN 5071C) by 2/434 bp (0.69%) in ITS, 10/693 bp (1.44%) in *RPB1*, 15/300 bp (5%, with five gaps (1%)) in *ACT*, and 23/460 bp (5%) in *CAL* and from *M. schoeni* (IRAN 5074C) by 7/682 bp (1.02%) in *RPB1*, 14/294 bp (4.76%, with six gaps (2%)) in *ACT*, and 26/460 bp (5.65%, with two gaps (0%)) in *CAL*. The PHI analysis further confirmed the absence of significant genetic recombination between *M. junci-acuti* and its closely related taxa (Φw > 0.05, Figure 4). *Macgarvieomyces junci-acuti* can be differentiated from *M. capsica* by having longer conidiophores (80–150 vs. 45–125 µm in *M. capsica*), longer conidia (25–33 vs. 22–27 µm in *M. capsica*), and the presence of hyphopodia, and from *M. schoeni* by having longer conidiophores (80–150 vs. 37–50 µm in *M. schoeni*).
Figure 6*Macgarvieomyces junci-acuti* (IRAN 4233C, ex-type). (**a**) Host (*Juncus* sp.); (**b**–**d**) colony on PDA (**b**), PCA (**c**), and MEA (**d**) after 14 days; (**e**,**f**) hyphopodia formed on SNA medium; (**g**–**i**) sporulation pattern on SNA medium ((**g**,**h**) = 10×, (**i**) = 40×); (**j**–**p**) conidiophores and conidia. Scale bars: (**e**,**f**,**j**–**p**) = 10 μm.
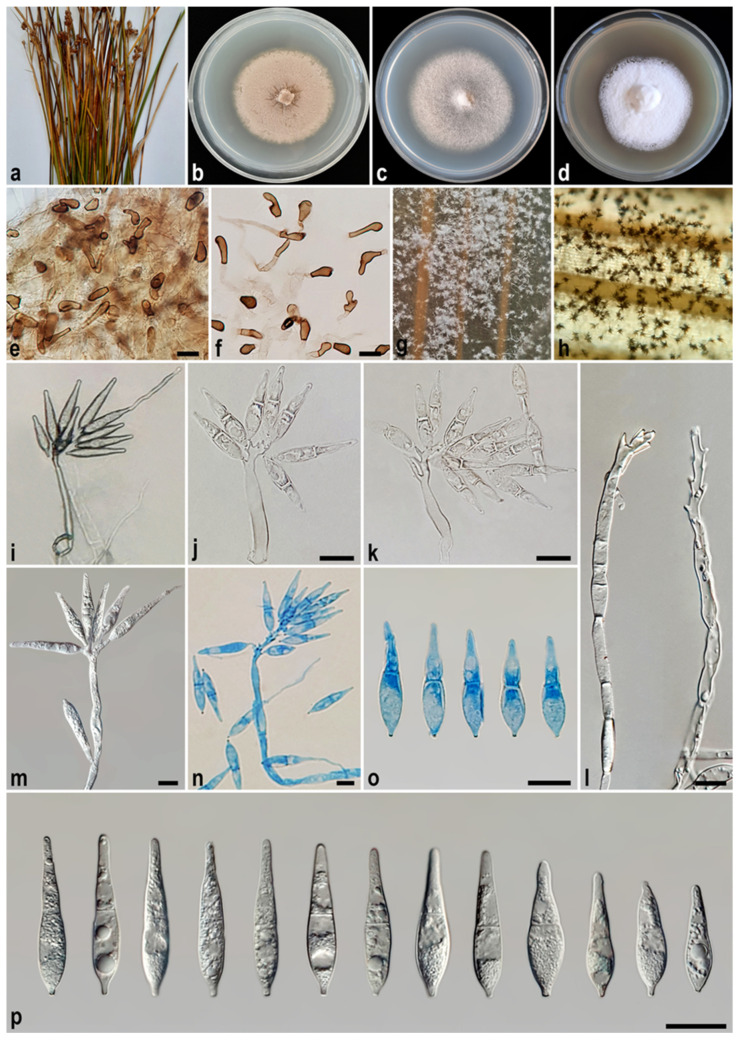



#### 3.2.4. *Macgarvieomyces juncigenus* A. Ahmadpour, Y. Ghosta, F. Alavi, Z. Alavi, and E. Hashemlou, sp. nov. (Figure 7)

MycoBank No. 859378

Etymology: Named after the host, *Juncus* sp., from which the holotype was collected.

Typification: Iran, Ardebil Province, Ardebil County, Fandoqlu Forest, isolated from the culms of *Juncus* sp. (*Juncaceae*, *Poales*), 10 June 2021, *A. Ahmadpour* (holotype IRAN 18497F, ex-type culture IRAN 5073C).

Description: *Asexual morph* on SNA medium with sterile barley seed: *Mycelium* consisting of smooth, hyaline, branched, septate hyphae, 1–2 μm diam. *Conidiophores* semi–macronematous, solitary, erect, straight–flexuous, smooth, mostly unbranched, occasionally branched, hyaline–pale brown, 1–3-septate, thick-walled near the base, occasionally swollen at the base, 35–75 × 5–7 µm ($\bar{x}$ = 49 × 6 μm, *n* = 50). *Conidiogenous cells* integrated, terminal, rarely intercalary, hyaline–pale brown, smooth, forming a rachis with several sympodially protruding flat-tipped denticles, 1–1.5 × 1–1.2 μm diam. *Conidia* solitary, narrowly obclavate–narrowly pyriform, hyaline, becoming pale brown with age, smooth, granular, guttulate, 1-septate, not or slightly constricted at septum, apex obtusely rounded, base tapering to a protruding hilum, 1–1.5 μm diam., not thickened, not darkened, 19–26 × 4–6 µm ($\bar{x}$ = 24.5 × 5.3 μm, *n* = 50). *Chlamydospores* 8–10 μm diam., formed in intercalary chains, spherical–ellipsoid, hyaline–pale brown, smooth, frequently giving rise to conidiophores. *Hyphopodia*, *sexual morph*, and *microconidiation* were not observed.

Culture characteristics: Colony on PDA reaching up to 30 and 49 mm diam. after 7 and 14 days at 25 °C in the dark, respectively; flat, circular, margin regular, velvety, white with buff centre and white aerial mycelium, reverse white with buff centre. Colony on PCA reaching up to 35 and 53 mm diam. after 7 and 14 days at 25 °C, respectively; flat, circular, margin regular, white–grey with white aerial mycelium, reverse grey at the centre and hyaline at the margin. Colony on MEA reaching up to 32 and 52 mm diam. after 7 and 14 days at 25 °C, respectively; flat, circular, margin entire, velvety, isabelline, or pale luteous with sparse white aerial mycelium, reverse ochreous–pale luteous towards the edge.

Additional specimen examined. Iran, Ardebil Province, Ardebil County, Fandoqlu Forest, isolated from the culms of *Juncus* sp. (*Juncaceae*, *Poales*), 10 June 2021, *A. Ahmadpour* (culture FCCUU 1961).

Notes: Phylogenetic analysis indicates a close relationship between *Macgarvieomyces juncigenus* and *M. salkadehensis*. A comparison of nucleotide differences in ITS, *RPB1*, *ACT*, and *CAL* indicates that *M. juncigenus* (IRAN 5073C) differs from *M. salkadehensis* (IRAN 5072C) by 1/481 bp (0.20%) in ITS, 2/653 bp (0.30%) in *RPB1*, 4/270 bp (1.48%) in *ACT*, and 4/462 bp (0.86%) in *CAL*. The PHI analysis results additionally verified that *M. juncigenus* does not exhibit significant genetic recombination with its closely related species (Φw > 0.05, Figure 4). Morphologically, *Macgarvieomyces juncigenus* differs from *M. salkadehensis* by having slightly longer and wider conidia (19–26 × 4–6 µm ($\bar{x}$ = 24.5 × 5.3 μm) vs. 15–28 × 4–5 µm ($\bar{x}$ = 22.5 × 4.8 μm) in *M. salkadehensis*); however, the overlapping morphological features and shared host between these two species complicate their distinction based solely on morphology. Consequently, molecular phylogenetic analyses are essential for accurately distinguishing *Macgarvieomyces* species and identifying any cryptic species.
Figure 7*Macgarvieomyces juncigenus* (IRAN 5073C, ex-type). (**a**) Host (*Juncus* sp.); (**b**–**d**) colony on PDA (**b**), PCA (**c**), and MEA (**d**) after 14 days; (**e**) sporulation pattern on SNA medium (10×); (**f**,**g**) chlamydospores formed on SNA medium; (**h**–**p**) conidiophores and conidia. Scale bars: (**f**–**p**) = 10 μm.
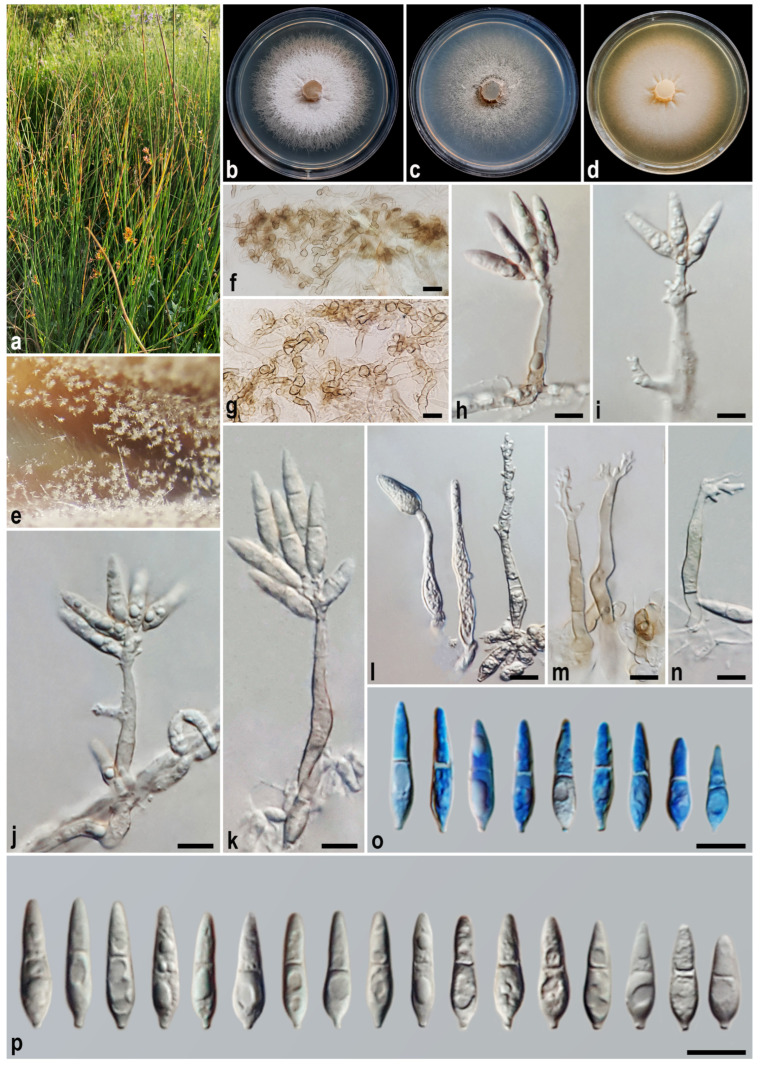



#### 3.2.5. *Macgarvieomyces salkadehensis* A. Ahmadpour, Y. Ghosta, F. Alavi, Z. Alavi, and E. Hashemlou, sp. nov. (Figure 8)

MycoBank No. 859379

Etymology: Named after the location, Salkadeh Village, Khoy County, from where the holotype was collected.

Typification: Iran, West Azarbaijan Province, Khoy County, Salkadeh Village, isolated from the culms of *Juncus* sp. (*Juncaceae*, *Poales*), 10 August 2021, *A. Ahmadpour* (holotype IRAN 18496F, ex-type culture IRAN 5072C).

Description: *Asexual morph* on SNA medium with sterile barley seed: *Mycelium* consisting of smooth, hyaline, branched, septate hyphae, 1–2 μm diam. *Conidiophores* semi–macronematous, solitary, erect, straight–flexuous, smooth, mostly unbranched, occasionally branched, hyaline–pale brown, 1–3-septate, thick-walled near the base, occasionally swollen at the base, 30–75 × 4–5 µm ($\bar{x}$ = 50 × 4.5 μm, *n* = 50). *Conidiogenous cells* integrated, terminal, rarely intercalary, hyaline–pale brown, smooth, forming a rachis with several sympodially protruding flat-tipped denticles, 1–2 × 1–1.5 μm. *Conidia* solitary, obclavate–narrowly obclavate, hyaline, becoming pale brown with age, smooth, granular, guttulate, 1-septate, not or slightly constricted at septum, apex obtusely rounded, base tapering to a protruding hilum, 1–1.5 μm diam., not thickened, not darkened, 15–28 × 4–5 µm ($\bar{x}$ = 22.5 × 4.8 μm, *n* = 50). *Chlamydospores* 6–10 μm diam, formed in intercalary chains, spherical–ellipsoid, hyaline–pale brown, smooth, frequently giving rise to conidiophores. *Hyphopodia*, *sexual morph*, and *microconidiation* were not observed.

Culture characteristics: Colony on PDA reaching up to 28 and 47 mm diam. after 7 and 14 days at 25 °C in the dark, respectively; flat, circular, margin entire, velvety, isabelline, or pale luteous with sparse white aerial mycelium, reverse ochreous–pale luteous towards the edge. Colony on PCA reaching up to 30 and 52 mm diam. after 7 and 14 days at 25 °C, respectively; flat, circular, margin regular, velvety, isabelline, or pale luteous with sparse white aerial mycelium, reverse ochreous–pale luteous towards the edge. Colony on MEA reaching up to 30 and 52 mm diam. after 7 and 14 days at 25 °C, respectively; flat, circular, margin entire, pale luteous without aerial mycelium, reverse pale luteous–hyaline towards the edge.

Additional specimens examined: Iran, Golestan Province, Gorgan County, Nahar Khoran Forest, Ziarat Village, isolated from the culms of *Juncus* sp. (*Juncaceae*, *Poales*), 4 November 2021, *A. Ahmadpour* (culture FCCUU 1957).—Iran, Ardebil Province, Ardebil County, Meshgin Shahr County, Razey City, isolated from the leaves and culms of *Scirpoides* sp. (*Cyperaceae*, *Poales*), 2 June 2022, *A. Ahmadpour* (culture FCCUU 1958).—*ibid*. on the culms of *Juncus* sp. (*Juncaceae*, *Poales*), 2 June 2022, *A. Ahmadpour* (culture FCCUU 1959).—Iran, Tehran Province, Damavand County, Haraz Road, isolated from the culms of *Juncus* sp. (*Juncaceae*, *Poales*), 20 June 2022, *E. Hashemlou* (culture FCCUU 1960).

Notes: Phylogenetic analyses (Figure 2) revealed that the five examined isolates of *Macgarvieomyces salkadehensis* formed a distinct clade, supported by 100% maximum likelihood (ML) bootstrap, 98% maximum parsimony (MP) bootstrap, and a Bayesian posterior probability (BI) of 0.93, and were closely related as sister taxa to *M. juncigenus*. A comparison of morphological characteristics, nucleotide sequence variations, and PHI analysis results (Φw > 0.05, Figure 4) for these species is provided in the notes section for *M. juncigenus*.
Figure 8*Macgarvieomyces salkadehensis* (IRAN 5072C, ex-type). (**a**) Host (*Juncus* sp.); (**b**–**d**) colony on PDA (**b**), PCA (**c**), and MEA (**d**) after 14 days; (**e**–**g**) sporulation pattern on SNA medium ((**e**,**g**) = 10×, (**f**) = 20×); (**h**) chlamydospores formed on SNA medium; (**i–s**) conidiophores and conidia. Scale bars: (**h**–**s**) = 10 μm.
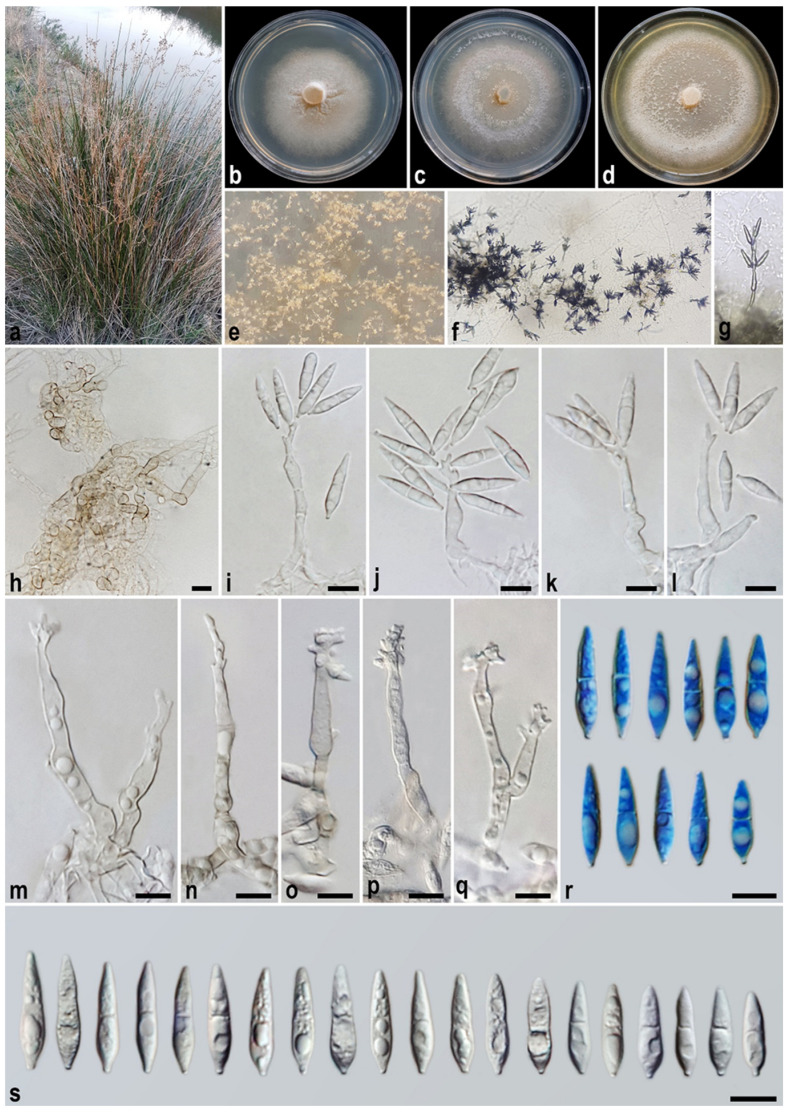



#### 3.2.6. *Macgarvieomyces schoeni* A. Ahmadpour, Y. Ghosta, F. Alavi, Z. Alavi, and E. Hashemlou, sp. nov. (Figure 9)

MycoBank No. 859380

Etymology: Named after the host, *Schoenus*, from which the holotype was collected.

Typification: Iran, West Azarbaijan Province, Khoy County, Salkadeh Village, isolated from the culms of *Schoenus* sp. (*Cyperaceae*, *Poales*), 11 September 2021, *A. Ahmadpour* (holotype IRAN 18498F, ex-type culture IRAN 5074C).

Description: *Asexual morph* on SNA medium with sterile barley seed: *Mycelium* consisting of smooth, hyaline, branched, septate hyphae, 1–2 μm diam. *Conidiophores* semi–macronematous, solitary, erect, straight–flexuous, smooth, unbranched, hyaline–pale brown, 1–3-septate, thick-walled near the base, 37–50 × 4–5 µm ($\bar{x}$ = 45 × 4.5 μm, *n* = 50). *Conidiogenous cells* integrated, terminal, rarely intercalary, hyaline–pale brown, smooth, forming a rachis with several sympodially protruding flat-tipped denticles, 1–1.5 × 1–1.2 μm. *Conidia* solitary, obclavate–narrowly obclavate, hyaline, becoming pale brown with age, smooth, granular, guttulate, 1-septate, not or slightly constricted at septum, apex obtusely rounded, base tapering to a protruding hilum, 1–1.5 μm diam., not thickened, not darkened, 26–32 × 4–5 µm ($\bar{x}$ = 29.5 × 4.3 μm, *n* = 50). *Hyphopodia* commonly formed, elongated, dome-shaped to multilobulate, brown–dark brown, smooth, 10–13 × 5–6 µm. *Sexual morph*, *microconidiation*, and *chlamydospores* were not observed.

Culture characteristics: Colony on PDA reaching up to 32 and 50 mm diam. after 7 and 14 days at 25 °C in the dark, respectively; flat, circular, margin entire, cottony appearance, white with white–grey aerial mycelium, reverse white–olivaceous grey towards the edge. Colony on PCA reaching up to 35 and 53 mm diam. after 7 and 14 days at 25 °C, respectively; flat, circular, margin regular, pale olivaceous grey with white aerial mycelium, reverse pale olivaceous grey at the centre and grey at the margin. Colony on MEA reaching up to 31 and 52 mm diam. after 7 and 14 days at 25 °C, respectively; flat, circular, margin entire, velvety, pale luteous with sparse white aerial mycelium; reverse ochreous–pale luteous towards the edge.

Additional specimen examined. Iran, West Azarbaijan Province, Khoy County, Salkadeh Village, isolated from the culms of *Schoenus* sp. (*Cyperaceae*, *Poales*), 11 September 2021, *A. Ahmadpour* (culture FCCUU 1962).

Notes: Phylogenetic analysis indicates that *Macgarvieomyces schoeni* is closely related to *M. caspica* and *M. junci-acuti* (Figure 2). Detailed comparisons of their morphological features, nucleotide variations, and PHI analysis results (Φw > 0.05, Figure 4) are provided in the notes sections for *M. caspica* and *M. junci-acuti*.
Figure 9*Macgarvieomyces schoeni* (IRAN 5074C, ex-type). (**a**,**b**) Symptoms on the culms of *Schoenus* sp.; (**c**–**e**) colony on PDA (**c**), PCA (**d**), and MEA (**e**) after 14 days; (**f**,**g**) hyphopodia formed on SNA medium; (**h**–**j**) sporulation pattern on SNA medium ((**h**) = 10×, (**i**,**j**) = 20×); (**k**–**n**) conidiophores and conidia. Scale bars: (**f**,**g**,**k**–**o**) = 10 μm.
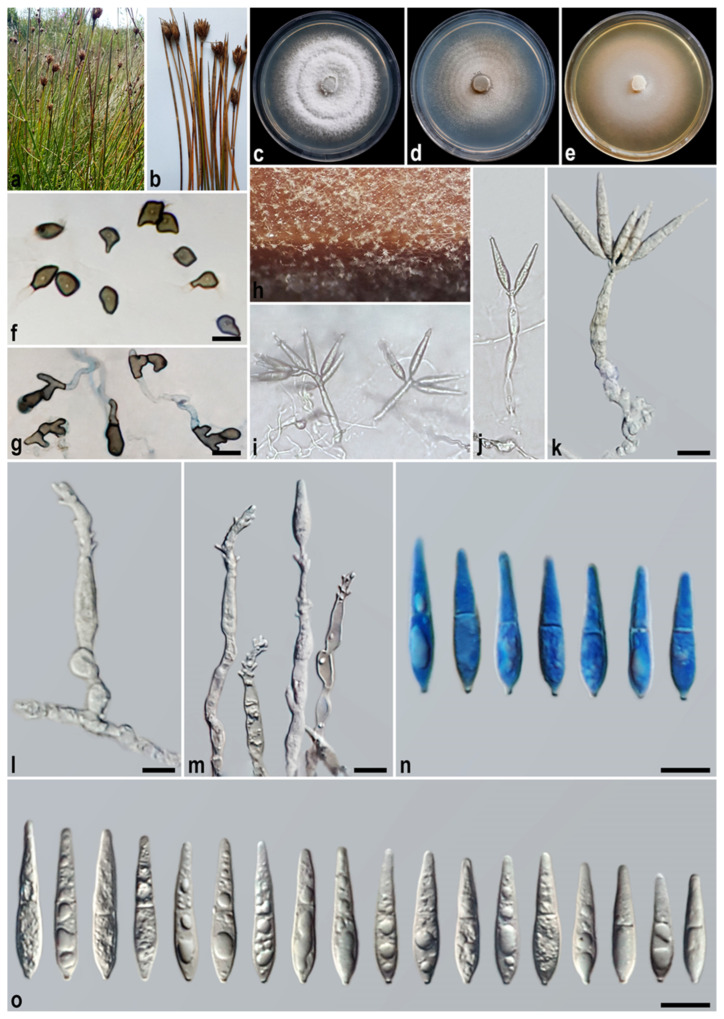



## 4. Discussion

This study assessed the species diversity of the genus *Macgarvieomyces* in Iran, focusing on host plants from the *Cyperaceae* and *Juncaceae* families. Our results reveal greater diversity within this genus than previously documented, leading to the identification and formal description of six new species. Until now, only three species had been described in this genus globally, all associated with hosts from these two plant families [2,5,9,15]. The newly discovered species in this study were also isolated exclusively from *Cyperaceae* and *Juncaceae*, suggesting a potential host-specific relationship. Interestingly, none of the previously described species, originally reported from Europe and New Zealand, were found in our samples. Among the newly described taxa, *Macgarvieomyces junci-acuti* and *M. salkadehensis* were isolated from multiple locations and host species within the same plant families, indicating a broader ecological distribution. In contrast, *M. caspica*, *M. cyperi*, *M. juncigenus*, and *M. schoeni*, were each isolated from a single host species and location. Despite these limitations in collection sites, the variety of habitats from which these fungi were obtained points to a wider ecological amplitude than previously recognized [2,9,15]. Further studies will be necessary to explore host specificity and geographical range within this genus.

Morphological features in *Macgarvieomyces*, as in other *Pyricularia*-like fungi, are often subtle and overlapping, making reliable identification based on morphology difficult, even at the generic level. Taxonomic features such as conidiophore structure, conidial shape and septation, pigmentation, and the presence of microconidia, hyphopodia, and chlamydospores show considerable variability and are often inconspicuous [2,5,9,34]. These challenges underscore the limitations of morphology-based classification and emphasize the necessity of molecular tools in fungal taxonomy. Recent studies using DNA-based phylogenetic analyses, particularly multi-locus sequence data, have greatly improved fungal taxonomy, resulting in the description of new families, genera, and species and have significantly helped in the understanding of evolutionary relationships within *Macgarvieomyces* and related genera [1,2,5,7,8,9,10]. The sequences of four genes, ITS, *RPB1*, *ACT*, and *CAL*, were used as DNA barcodes to differentiate *Macgarvieomyces* spp. Our study employed sequences from four genetic markers, ITS, *RPB1*, *ACT*, and *CAL*, as DNA barcodes to resolve species boundaries and infer phylogenetic relationships within *Macgarvieomyces*. Notably, *M. salkadehensis* and *M. juncigenus* shared overlapping morphological traits that made them difficult to distinguish based on morphology alone; however, phylogenetic analyses successfully differentiated them.

As part of our broader investigation into the fungal diversity of Iranian wetlands, we have recovered numerous fungal taxa from hosts in the *Cyperaceae* and *Juncaceae* families [17,18,35,36,37]. These taxa include fungi with varied ecological roles, ranging from pathogens and saprophytes to endophytes. The discovery of new *Macgarvieomyces* species in this study reinforces the ecological richness and taxonomic novelty of fungi associated with Iran’s wetland habitats. It also highlights the value of under-studied ecosystems and plant-fungal associations for uncovering previously unknown fungal taxa. Given Iran’s unique geographic position and environmental diversity, it will likely harbour additional, undescribed fungal species, especially among ecologically specialized genera like *Macgarvieomyces*. Continued exploration of these habitats, guided by integrative taxonomic approaches that combine morphological and molecular data, is essential for revealing hidden fungal diversity and informing future conservation and ecological efforts.

## 5. Conclusions

This study makes a substantial contribution to the understanding of *Macgarvieomyces* species diversity in Iran by identifying and describing six novel species, collected from wetland habitats and host plants belonging to the *Cyperaceae* and *Juncaceae* families. By integrating morphological features with multi-locus phylogenetic analyses of DNA barcodes, we present robust evidence supporting the delineation of distinct species. Our results not only broaden the taxonomic scope of the genus *Macgarvieomyces* but also highlight the ecological significance of Iranian wetlands as vital reservoirs of both known and previously undocumented fungal diversity. Continued research and conservation of these ecosystems are crucial for protecting fungal biodiversity and discovering species with potential applications in agriculture, biotechnology, and environmental monitoring. This study provides a foundation for future taxonomic, ecological, and functional research on fungi in these largely unexplored regions.

## Figures and Tables

**Figure 1 jof-11-00489-f001:**
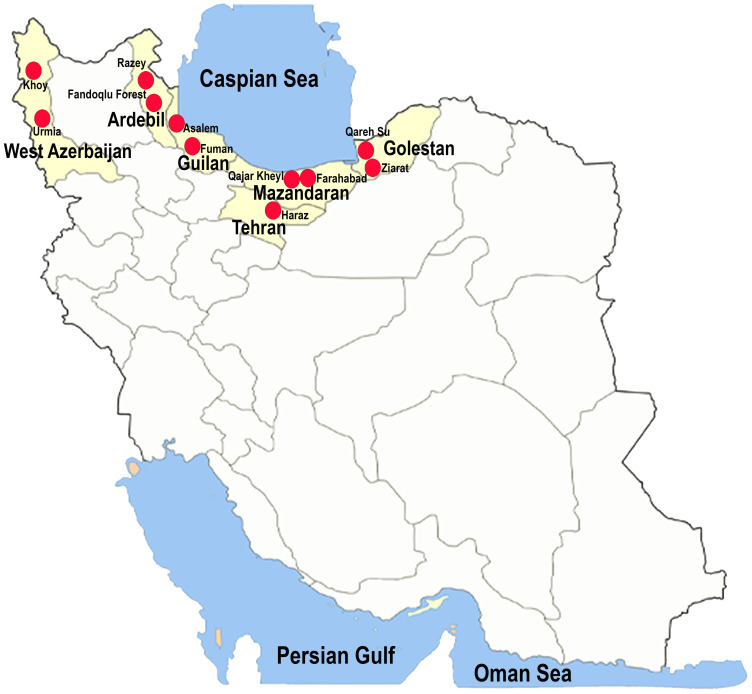
Map of Iran indicating the sampling locations across the provinces of Ardebil, Golestan, Guilan, Mazandaran, Tehran, and West in the present study.

**Figure 2 jof-11-00489-f002:**
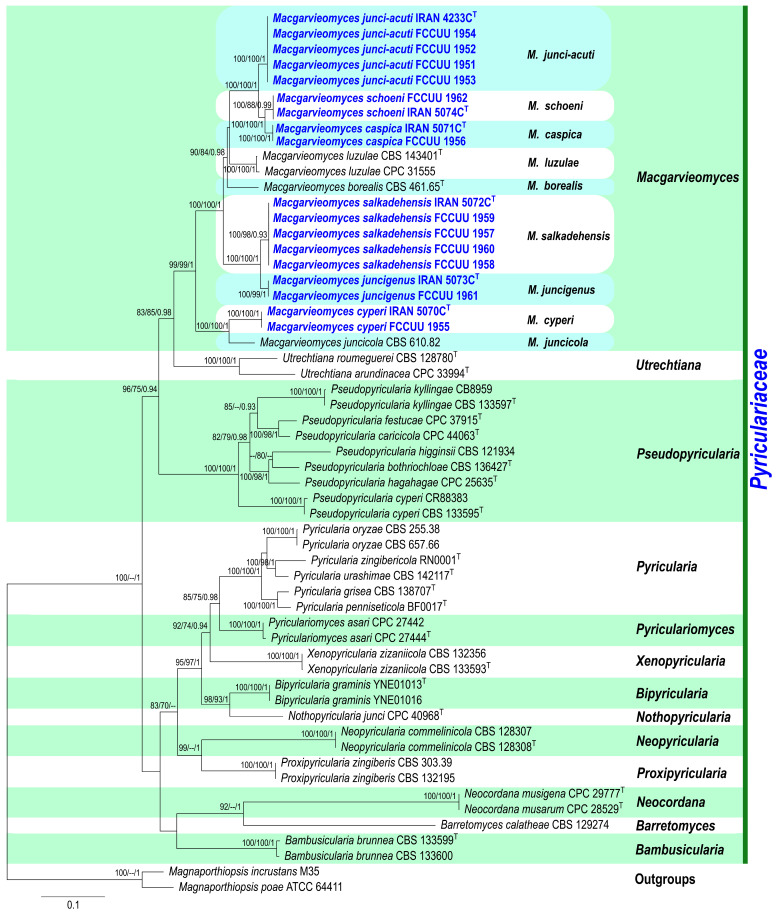
Phylogenetic tree constructed using maximum likelihood (ML) analysis based on the combined ITS, *RPB1*, *ACT*, and *CAL* sequence data from *Macgarvieomyces* species and related genera within the *Pyriculariaceae* family. Support values at each node include bootstrap percentages from both maximum likelihood and maximum parsimony (MLBS/MPBS) analyses equal to or greater than 70%, as well as Bayesian posterior probabilities (BIPP) of 0.90 or higher. The newly identified strains are highlighted in bold blue, with the phylogenetic tree rooted using *Magnaporthiopsis incrustans* (M35) and *Ma. poae* (ATCC 64411). The number of substitutions for nucleotides is displayed on the scale bar and ex-type strains are denoted by ^T^.

**Table 1 jof-11-00489-t001:** Information of taxa and corresponding GenBank accession numbers used for phylogenetic analyses. The sequences generated in this study are in bold, ^T^ indicates ex-type strains, and “-“ indicates that data were unavailable.

Species	Culture Collection Number	Host/Substrate	Location	GenBank Accession Numbers			
				ITS	*RPB1*	*ACT*	*CAL*
*Bambusicularia brunnea*	CBS 133599 ^T^	*Sasa* sp.	Japan	KM484830	KM485043	AB274449	AB274482
*Bambusicularia brunnea*	CBS 133600	*Phyllostachys bambusoides*	Japan	AB274436	KM485044	AB274450	AB274483
*Barretomyces calatheae*	CBS 129274	*Calathea longifolia*	Brazil	KM484831	KM485045	KM485162	KM485231
*Bipyricularia graminis*	YNE01013 ^T^	*Poaceae* sp.	China	MW479090	MW482852	OQ918100	-
*Bipyricularia graminis*	YNE01016	*Poaceae* sp.	China	MW479091	MW482853	OQ918101	-
*Macgarvieomyces borealis*	CBS 461.65 ^T^	*Juncus effusus*	Scotland	KM484854	KM485070	KM485170	KM485239
** *Macgarvieomyces caspica* **	**IRAN 5071C** ^T^	** *Juncus acutus* **	**Iran**	**PQ453843**	**PQ450660**	**PQ450624**	**PQ450642**
** *Macgarvieomyces caspica* **	**FCCUU 1956**	** *Juncus acutus* **	**Iran**	**PQ453844**	**PQ450661**	**PQ450625**	**PQ450643**
** *Macgarvieomyces cyperi* **	**IRAN 5070C** ^T^	** *Cyperus* ** **sp.**	**Iran**	**PQ453841**	**PQ450658**	**PQ450622**	**PQ450640**
** *Macgarvieomyces cyperi* **	**FCCUU** **1955**	** *Cyperus* ** **sp.**	**Iran**	**PQ453842**	**PQ450659**	**PQ450623**	**PQ450641**
** *Macgarvieomyces junci-acuti* **	**IRAN 4233C** ^T^	** *Juncus acutus* **	**Iran**	**PQ453836**	**PQ450653**	**PQ450617**	**PQ450635**
** *Macgarvieomyces junci-acuti* **	**FCCUU 1951**	** *Juncus* ** **sp.**	**Iran**	**PQ453837**	**PQ450654**	**PQ450618**	**PQ450636**
** *Macgarvieomyces junci-acuti* **	**FCCUU 1952**	** *Juncus* ** **sp.**	**Iran**	**PQ453838**	**PQ450655**	**PQ450619**	**PQ450637**
** *Macgarvieomyces junci-acuti* **	**FCCUU 1953**	** *Juncus* ** **sp.**	**Iran**	**PQ453839**	**PQ450656**	**PQ450620**	**PQ450638**
** *Macgarvieomyces junci-acuti* **	**FCCUU 1954**	** *Schoenus* ** **sp.**	**Iran**	**PQ453840**	**PQ450657**	**PQ450621**	**PQ450639**
*Macgarvieomyces juncicola*	CBS 610.82	*Juncus effusus*	Netherlands	KM484855	KM485071	KM485171	KM485240
** *Macgarvieomyces juncigenus* **	**IRAN 5073C** ^T^	** *Juncus* ** **sp.**	**Iran**	**PQ453850**	**PQ450667**	**PQ450631**	**PQ450649**
** *Macgarvieomyces juncigenus* **	**FCCUU 1961**	** *Juncus* ** **sp.**	**Iran**	**PQ453851**	**PQ450668**	**PQ450632**	**PQ450650**
*Macgarvieomyces luzulae*	CBS 143401 ^T^	*Luzula sylvatica*	Ukraine	MG934440	MG934469	MG934462	MG934519
*Macgarvieomyces luzulae*	CPC 31555	*Luzula sylvatica*	Ukraine	MG934441	MG934470	MG934463	MG934520
** *Macgarvieomyces salkadehensis* **	**IRAN 5072C** ^T^	** *Juncus inflexus* **	**Iran**	**PQ453845**	**PQ450662**	**PQ450626**	**PQ450644**
** *Macgarvieomyces salkadehensis* **	**FCCUU 1957**	** *Juncus* ** **sp.**	**Iran**	**PQ453846**	**PQ450663**	**PQ450627**	**PQ450645**
** *Macgarvieomyces salkadehensis* **	**FCCUU 1958**	** *Scirpoides* ** **sp.**	**Iran**	**PQ453847**	**PQ450664**	**PQ450628**	**PQ450646**
** *Macgarvieomyces salkadehensis* **	**FCCUU 1959**	** *Juncus* ** **sp.**	**Iran**	**PQ453848**	**PQ450665**	**PQ450629**	**PQ450647**
** *Macgarvieomyces salkadehensis* **	**FCCUU 1960**	** *Juncus* ** **sp.**	**Iran**	**PQ453849**	**PQ450666**	**PQ450630**	**PQ450648**
** *Macgarvieomyces schoeni* **	**IRAN 5074C** ^T^	** *Schoenus* ** **sp.**	**Iran**	**PQ453852**	**PQ450669**	**PQ450633**	**PQ450651**
** *Macgarvieomyces schoeni* **	**FCCUU** **1962**	** *Schoenus* ** **sp.**	**Iran**	**PQ453853**	**PQ450670**	**PQ450634**	**PQ450652**
*Magnaporthiopsis incrustans*	M35	*-*	-	JF414843	Genome	Genome	Genome
*Magnaporthiopsis poae*	ATCC 64411	*Triticum* sp.	USA	Genome	Genome	AF395973	AF396032
*Neocordana musarum*	CBS 142116 ^T^	*Musa* sp.	France	KY173425	KY173577	KY173568	-
*Neocordana musigena*	CBS 142624 ^T^	*Musa* sp.	Morocco	KY979749	KY979886	KY979855	-
*Neopyricularia commelinicola*	CBS 128307	*Commelina communis*	South Korea	FJ850125	KM485086	KM485174	KM485243
*Neopyricularia commelinicola*	CBS 128308 ^T^	*Commelina communis*	South Korea	FJ850122	KM485087	KM485175	-
*Nothopyricularia junci*	CBS 148308 ^T^	*Juncus effusus*	Netherlands	OK664720	OK651152	OK651127	OK651142
*Proxipyricularia zingiberis*	CBS 132195	*Zingiber mioga*	Japan	KM484869	KM485088	AB274448	KM485244
*Proxipyricularia zingiberis*	CBS 303.39	*Zingiber officinale*	Japan	KM484871	KM485092	KM485177	KM485247
*Pseudopyricularia bothriochloae*	CBS 136427 ^T^	*Bothriochloa bladhii*	Thailand	KF777186	KY905701	KY905700	-
*Pseudopyricularia caricicola*	CBS 149674 ^T^	*Carex disticha*	Netherlands	OQ628482	-	OQ627932	-
*Pseudopyricularia cyperi*	CBS 133595 ^T^	*Cyperus iria*	Japan	KM484872	AB818013	AB274453	AB274485
*Pseudopyricularia cyperi*	Cr88383	*Cyperus rotundus*	Philippines	KM484874	KM485094	KM485179	KM485249
*Pseudopyricularia festucae*	CBS 146629 ^T^	*Festuca californica*	USA	MW883447	MW890057	-	MW890044
*Pseudopyricularia hagahagae*	CPC 25635 ^T^	Unidentified *Cyperaceae*	South Africa	KT950851	KT950877	KT950873	-
*Pseudopyricularia higginsii*	CBS 121934	*Typha orientalis*	New Zealand	KM484875	KM485095	KM485180	KM485250
*Pseudopyricularia kyllingae*	CBS 133597 ^T^	*Kyllinga brevifolia*	Japan	KM484876	KM485096	AB274451	AB274484
*Pseudopyricularia kyllingae*	PH0054 = Cb8959	*Cyperus brevifolius*	Philippines	KM484877	KM485097	KM485181	KM485251
*Pyricularia grisea*	CBS 138707 ^T^	*Digitaria* sp.	USA	KM484885	KM485105	KM485187	KM485258
*Pyricularia oryzae*	CBS 255.38	-	Romania	KM484889	KM485109	KM485190	KM485261
*Pyricularia oryzae*	CBS 657.66	*Oryza sativa*	Egypt	KM484893	KM485113	KM485194	KM485265
*Pyricularia penniseticola*	BF0017	*Pennisetum typhoides*	Burkina Faso	KM484925	KM485144	DQ240878	DQ240894
*Pyricularia urashimae*	CBS 142117 ^T^	*Urochloa brizantha*	Brazil	KY173437	KY173578	KY173571	KX524100
*Pyricularia zingibericola*	RN0001 ^T^	*Zingiber officinale*	Réunion	KM484941	KM485157	KM485157	KM485297
*Pyriculariomyces asari*	CPC 27442	*Asarum* sp.	Malaysia	KX228290	MG934472	KX228360	-
*Pyriculariomyces asari*	CPC 27444 ^T^	*Asarum* sp.	Malaysia	KX228291	KX228368	KX228361	MG934541
*Utrechtiana arundinacea*	CPC 33994 ^T^	*Phragmites* sp.	Netherlands	MG934461	MG934473	MG934468	MG934542
*Utrechtiana roumeguerei*	CBS 128780 ^T^	*Phragmites australis*	Netherlands	JF951153	KM485047	KM485163	KM485232
*Xenopyricularia zizaniicola*	CBS 133593 ^T^	*Zizania latifolia*	Japan	KM484947	KM485161	KM485230	AB274479
*Xenopyricularia zizaniicola*	CBS 132356	*Zizania latifolia*	Japan	KM484946	KM485160	AB274444	AB274480

**Table 2 jof-11-00489-t002:** Phylogenetic details of individual and concatenated sequence datasets utilized in phylogenetic analyses.

Parameter	Gene				
	ITS	*RPB1*	*ACT*	*CAL*	Combined
Number of taxa	57	56	56	48	57
Total characters	517	735	513	670	2435
Constant sites	327	382	224	252	1185
Variable sites	190	353	289	418	1250
Parsimony informative sites	162	329	258	372	1121
Parsimony uninformative sites	28	24	31	46	129
AIC substitution model *	GTR+I+G	GTR+I+G	HKY+I+G	HKY+I+G	GTR+I+G
Lset nst, Rates	6, invgamma	6, invgamma	2, invgamma	2, invgamma	6, invgamma
−lnL	4564.091206	6165.249261	5332.845681	7571.390091	23503.286101

* Substitution models selected based on the Akaike Information Criterion (AIC) and applied in Bayesian inference analyses.

## Data Availability

The original contributions presented in this study are included in the article. Further inquiries can be directed to the corresponding author.

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
