# Peer review of "Additions to Macgarvieomyces in Iran: Morphological and Phylogenetic Analyses Reveal Six New Species"

_jof, 2025, doi:10.3390/jof11070489_

Round 1
Reviewer 1 Report
The attached PDF file of the manuscript contains some corrections and suggestions.
The attached PDF file of the manuscript contains some corrections and suggestions.

Author Response
The authors sincerely thank the esteemed reviewers for their thorough evaluation, insightful suggestions, and constructive comments, which significantly enhanced the quality of the manuscript. We have carefully reviewed the manuscript and implemented all the recommended revisions, along with several additional minor corrections we identified independently; all changes are highlighted in yellow. Furthermore, the manuscript text has been revised to reduce redundancy, in accordance with the attached duplication report, with the corresponding modifications also marked in yellow. Figures and references have been updated and corrected throughout the text of the manuscript. The names of two species (M. juncigenus and M. schoeni) were revised in accordance with MycoBank records. These modifications have been applied and highlighted in yellow throughout the figures, tables, and main text of the manuscript. A detailed, point-by-point response to each reviewer comment is provided below.
With kind regards,
Reviewer 1:
Q/R1- The title proposed to change as: Six New Macgarvieomyces Species from Iran Uncovered by Morphological and Phylogenetic Evidence.
Answer: We think the title is clear and indicative, and added a large to the Macgarvieomyces taxonomy, so it was not changed.
Q/Rs-
Line 18: delete: formerly placed in the genus Pyricularia.
Answer: It was deleted.
Line 19: change the word from to only in.
Answer: It was changed as the previously known species are reported only in Europe and New Zealand.
Lines 22 and 23: Replace — instead:
Answer: It was replaced.
Line 24: Replace examination with characterization:
Answer: Here, examination is better than characterization, as in examination, all morphological features are characterized. So, it was not changed.
Line 24: add a before multi-locus…
Answer: It was not added as we used analyses, a plural form of analysis.
Line 26. Replace preferences with distribution
Answer: The ecological preferences are better than the distribution, so it was not changed.
Line 36: added words class and phylum.
Answer: This is a norm in most papers, and it was not changed.
Line 43: Change conidial features in asexual… to the morphological characteristics of the.
Answer: It was changed.
Line 45, at what level
Answer: As it was cited in the first part of the paragraph, it was referred to all taxonomic levels below the taxon order.
Line 49: I or II?
Answer: II is correct.
Line 51: complete or fragments?
Answer: parts of these genes, so the word parts was added at the first of line 43.
Line 53: Put in alphabetical order.
Answer: It was corrected.
Line 55: read line 40.
Answer: The sentences was deleted, as it was repeated in line 40.
Line 59: delete a sentence.
Answer: It was deleted.
Line 66: delete Cereals.
Answer: It was deleted.
Line 85: Please, include a map of Iran with the sampled sites.
Answer: we included Iran map with sampling locations (Figure 1).
Lines 92, 93: This is acceptable, but I strongly recommend to the authors deposit the ex-type strains in an international, more accessible culture collection.
Answer: We are very interested in depositing our isolates in more than one international collection, but at present, we have limitations in doing that, so we deposited the isolates only in our internationally accessible collection. Iran cultures (IRAN…F, IRAN…C) can be seen on the site https://gcm.wdcm.org/ and is usable and accessible to researchers.
Line 96: include the medium formula.
Answer: We used the commercially prepared PDA medium, as we wrote. We did not use hand-made medium, including the extracts of 200-250 g white potato, 20 g D-glucose, 20 g Agar, and 1000 ml distilled water.
Line 98: include the medium formula.
Answer: As the PDA medium, we used the commercially prepared MEA medium.
Line 103: include the medium formula.
Answer: It was included.
Line 112: correct the words.
Answer: It was corrected.
Line 115: Are these authors whose that described these methodology at the first time? Please, try to not use self-citations.
Answer: No, these authors did not describe the method for the first time, but they used the same method described in the references. To avoid self-citation, the ref. 17 was deleted.
Line 159: word corrections.
Answer: The mentioned parts were corrected
Line 216. Please, improve the image quality.
Answer: The figure we sent had good quality. Here, the quality was poor. We will attach a good-quality figure to the manuscript.
Line 270 and others about the quality of fungal figures.
Answer: We checked the quality of the figures, and will improve their quality as well as possible.

Reviewer 2 Report
Nicely written article, with professionally supported results. Agree with Authors: "This work significantly enhances the known diversity of Macgarvieomyces"
The new strains (at least the type starins) sholud be deposited in at least 2 different WDCM registered culture collections, in order to make them accessible to the scientific community.
It would greatly facilitate a thorough review if the new sequences used and deposited were available. Unfortunately, these are not yet released in GenBank. If at least the ITS sequences of the type strains of the new species could be available, that would be a great help.
Quality (resolution) of Fig 1 should be enhanced.
Author Response
The authors sincerely thank the esteemed reviewers for their thorough evaluation, insightful suggestions, and constructive comments, which significantly enhanced the quality of the manuscript. We have carefully reviewed the manuscript and implemented all the recommended revisions, along with several additional minor corrections we identified independently; all changes are highlighted in yellow. Furthermore, the manuscript text has been revised to reduce redundancy, in accordance with the attached duplication report, with the corresponding modifications also marked in yellow. Figures and references have been updated and corrected throughout the text of the manuscript. The names of two species (M. juncigenus and M. schoeni) were revised in accordance with MycoBank records. These modifications have been applied and highlighted in yellow throughout the figures, tables, and main text of the manuscript. A detailed, point-by-point response to each reviewer comment is provided below.
With kind regards,
Reviewer 2:
Q/R1- The new strains (at least the type strains) should be deposited in at least 2 different WDCM registered culture collections, in order to make them accessible to the scientific community.
Answer: As it was cited above, we are very interested in depositing our isolates in other international collections, but due to our limitations, we could not do that now. We will deposit all of our isolates in other international collections once our limitations are removed. Iran cultures (IRAN…F, IRAN…C) can be seen on the site https://gcm.wdcm.org/ and is usable and accessible to researchers.
Q/R2- It would greatly facilitate a thorough review if the new sequences used and deposited were available. Unfortunately, these are not yet released in GenBank. If at least the ITS sequences of the type strains of the new species could be available, that would be a great help.
Answer: The new sequences will be released immediately after the final acceptance and publication of the manuscript.
Q/R3- Quality (resolution) of Fig 1 should be enhanced.
Answer: It will be improved in the best form.
